# Survey of Distributed and Decentralized IoT Securities: Approaches Using Deep Learning and Blockchain Technology

Ayodeji Falayi , Qianlong Wang * , Weixian Liao  and Wei Yu

Department of Computer and Information Sciences, Towson University, Towson, MD 21252, USA;
afalay1@students.towson.edu (A.F.); wliao@towson.edu (W.L.); wyu@towson.edu (W.Y.)
* Correspondence: qwang@towson.edu

**Abstract:** The Internet of Things (IoT) continues to attract attention in the context of computational resource growth. Various disciplines and fields have begun to employ IoT integration technologies in order to enable smart applications. The main difficulty in supporting industrial development in this scenario involves potential risk or malicious activities occurring in the network. However, there are tensions that are difficult to overcome at this stage in the development of IoT technology. In this situation, the future of security architecture development will involve enabling automatic and smart protection systems. Due to the vulnerability of current IoT devices, it is insufficient to ensure system security by implementing only traditional security tools such as encryption and access control. Deep learning and blockchain technology has now become crucial, as it provides distinct and secure approaches to IoT network security. The aim of this survey paper is to elaborate on the application of deep learning and blockchain technology in the IoT to ensure secure utility. We first provide an introduction to the IoT, deep learning, and blockchain technology, as well as a discussion of their respective security features. We then outline the main obstacles and problems of trusted IoT and how blockchain and deep learning may be able to help. Next, we present the future challenges in integrating deep learning and blockchain technology into the IoT. Finally, as a demonstration of the value of blockchain in establishing trust, we provide a comparison between conventional trust management methods and those based on blockchain.

**Keywords:** deep learning; Internet of Things security; distributed and decentralized system

## 1. Introduction

The Internet of Things (IoT) continues to attract attention, as it can effectively and intelligently sense the environment through a series of smart devices and enable various smart applications [1,2]. In particular, the IoT has been developed to enable smart applications, such as smart cities, smart transportation, smart homes, smart vehicles, smart hospitals, etc. [3,4]. It is expected that by 2026, more than 100 billion smart devices will be deployed, which is unprecedented in computer history [5]. However, there are also a variety of security concerns related to the enormous quantity of data streams transferring among smart devices in the context of the deployment of IoT technologies [6]. The application of traditional security protection measures, such as encryption, authentication, and access control, has been deemed insufficient for systems with a large number of linked devices in which each device them is subject to particular vulnerabilities [7,8]. For instance, the security methodologies used in current IoT technologies and protocols, such as MQTT (Message Queue Telemetry Transport), Z-Wave, ZigBee, RFID (radio frequency identification), etc., have become insufficient to warrant full confidence in their reliability, demonstrating that such security mechanisms are likely to be defeated by new forms of attacks that cannot be detected by existing solutions. Furthermore, identifying which solutions are acceptable for securing IoT systems is difficult because of the challenges brought about by the emergence of a wide range of heterogeneous and decentralized IoT devices [7].

As a tremendous amount of data is being created by a variety of IoT applications, deep learning has now become an effective solution for IoT security issues. Deep learning is a well-known machine learning model built by artificial neural networks (ANNs) that aims to mimic the human nervous systems by using layers of perceptrons [9]. ANNs have been widely studied in recent years, as computational power has dramatically increased and shown a strong ability to perform various tasks, e.g., image recognition, event prediction, fault detection, etc. [10–12]. Typically, an ANN structure contains a number of layers, each of which contains a number of nodes or "neurons". Such nodes are assigned a learned weight and can be stimulated by certain inputs. It has been proven that with enough layers and nodes, an ANN is able to mine complex patterns among large-scale dataset. The application of deep learning to IoT security is currently a new and active research field that is being widely studied.

On the other hand, most existing security methods are based on a centralized system architecture, which may not be suitable for large-scale IoT systems. One breach in such a security design might leave the entire network vulnerable. Many of the security parameters used in IoT devices rely on cryptographic algorithms that demand substantial resources [13]. Furthermore, since heterogeneous IoT system platforms have been studied, it is challenging to build one security protocol that is compatible across all platforms. To address the security issue in decentralized IoT systems, blockchain technology has been studied and adopted to build secure IoT applications [14,15]. Transaction authentication on a blockchain does not require the approval of a central authority, as is the case in centralized systems. Instead, it relies on the approval of a number of participants within the system. Accordingly, blockchain technology can reliably and transparently record transactions reliably owing to its decentralized architecture. Considering the decentralized paradigm of IoT systems, blockchain has emerged as a promising solution to ensure IoT security [16].

In general, deep learning and blockchain have become the most popular and promising techniques for the development of secure and decentralized IoT systems. Furthermore, researchers currently realize that these two technologies may have a complementary relationship when combined to implement a fully secure and smart IoT system. On one hand, the capacity of deep learning for intelligent data analysis and decision making could make blockchain applications more effective and feasible. On the other hand, blockchain has the potential to aid deep learning by providing a large amount of data in a decentralized manner.

To the best of our knowledge, no literature survey of the application of both deep learning and blockchain techniques for IoT security has been conducted to date. In this work, we summarize the existing works on the application of deep learning and blockchain to enable secure decentralized IoT systems. Furthermore, we discuss the importance and future challenges of integrating deep learning and blockchain in IoT systems.

The main findings of this paper are summarized as follows.

- We provide a literature survey of security methods for the IoT, including deep learning and blockchain. We also provide a summary of the primary benefits and drawbacks of various methods.
- We discuss how deep learning and blockchain technology can overcome the current difficulties associated with ensuring the safety and reliability of IoT devices.
- We highlight the contrast between decentralized blockchain-based security solutions and deep learning methods.

The rest of this survey paper is organized as follows. In Section 2, we present an overview of the scope of this survey. We detail the current deep learning IoT security solutions and blockchain applications in IoT security in Sections 3 and 4, respectively. In Section 5, we present a discussion and explored future challenges associated with deploying deep learning and blockchain for IoT security.

## 2. Overview of IoT Security

In this section, we review IoT security in detail, including its architecture, security challenges, and requirements. We also review machine learning and blockchain applications in IoT systems.

### *2.1. Internet of Things*

2.1.1. IoT Architecture

The IoT refers to a collection of interconnected smart devices that may share and receive data via wired or wireless networks [17]. Typically, the IoT is considered a system of real-world devices, e.g., sensors, cameras, routers, etc. Some examples of IoT are shown in Figure 1. Not only are these examples integrated with software that allows them to receive, evaluate, and share data with other devices, but they also use such data to drive other decisions.

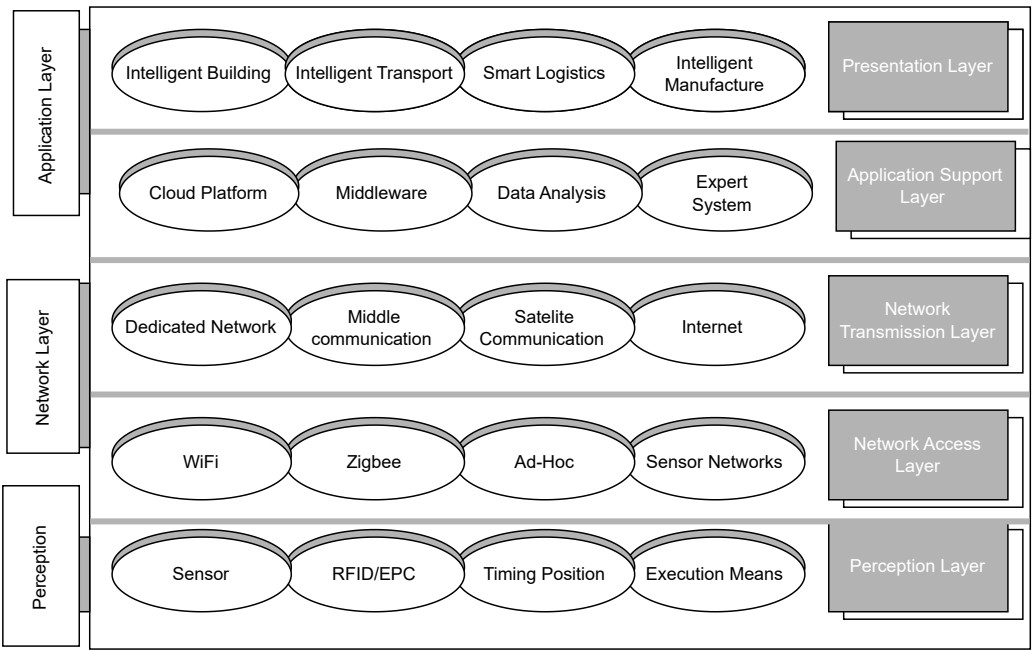

**Figure 1.** Overview of IoT architecture.

The majority of studies have outlined a three-tiered structure for the IoT, which includes perception, network, and application layers, as shown in Figure 1). Typically, the perception layer refers to smart edge devices (e.g., sensors, cameras, etc.), which can sense or record the environment and generate and process data. In particular, the data from the devices are converted to a digital format in this layer. This layer also picks up on variations in the sensing environment's physical parameters in near-real time. Some common threats in this layer include eavesdropping, replay attacks, and timing attacks [18]. The network layer is used to establish communications among individual smart devices. This layer transfers data from the perception layer to the application layer. Due to its critical nature, the transmission of genuine and unaltered data represents a significant problem in this layer. The application layer, on the other hand, is used to develop applications to enable the use of certain IoT services by users. The functionality of an IoT network is described in Figure 1. The gateway acts as a conduit for the movement of information between physical devices (such as sensors, cameras, etc.) and the application layer. Information gathered in the perception layer can be sent to end users via the gateway or the Internet.

2.1.2. IoT Security Challenges

The nature of the interconnected structure of IoT systems is one of the major reasons causes of risks of malicious attacks [19]. As shown in Figure 2, specific attacks are designed

for each layer of the IoT architecture. According to the literature, the typical method for ensuring security is ineffective, especially when it comes to user safety in IoT systems [20]. In addition, some deliberately designed attacks are able to not only compromise certain devices in the IoT but also cause a series of failures as a a consequence, i.e., cascading failures [21]. Therefore, it is important to secure every processing phase of the IoT before data are fed to a higher layer [22].

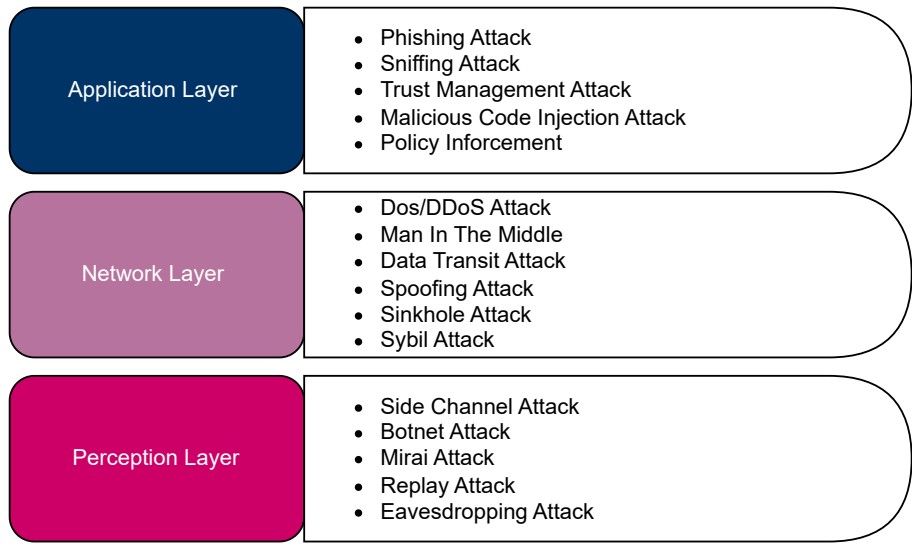

**Figure 2.** Attacks on IoT layers.

On the other hand, most existing IoT applications assume that there is a central server that can remotely communicate and control the devices. Due to the pressure to quickly launch applications in the market, security issues may be overlooked. For instance, the software system of the applications must be improved, in addition to the security of lower levels, such as hardware and firmware. Many devices in the IoT, on the other hand, are not updated over time, which may lead to system vulnerability [23].

As a result, there is an urgent need for critical attention to current security concepts regarding effective IoT security procedures. The main reasons for this urgency can be summarized as follows. (i) With booming data and devices, various novel IoT applications are explored promptly. Many of these applications may not have been systematically analyzed for potential risks of malicious attacks and lack a proper design that is resilient to system attacks. (ii) Conventional security approaches such as encryption, authentication mechanisms, access control algorithms, etc., are becoming less effective in large-scale IoT scenarios [24]. Novel technology may also enable novel attacks that are able to bypass traditional security approaches. Furthermore, novel attacks may be able to attack some parts of IoT applications that are not protected by traditional security measures. (iii) Most existing IoT systems utilize a wide range of dependable and cost-effective smart devices. Such devices are normally resource-limited. Hence, it becomes even more difficult to implement security algorithms to detect or eliminate attacks [25].

2.1.3. Security Requirements for IoT

To achieve successful security methods, the following characteristics should be considered. The important security requirements are shown in Figure 3, which include confidentiality, integrity, authentication, availability, non-repudiation, and authorization [21].

- *Confidentiality:* Sensitive and private information and data should never be disclosed or inferred by malicious users [25];
- *Integrity:* As data are acquired, they should never be tampered with by an unauthorized user, especially if the communication is launched over an unsecured network [26];

- *Authentication:* The transmission and processing of the data should be able to be verified following designed protocols in the IoT system [27];
- *Authorization:* Only users that are granted authorization should be able to access the IoT system and data [28];
- *Availability:* All authorized users must have access to the services transmitted by IoT systems. A compelling configuration of IoT systems should prioritize availability over all other properties [29];
- *Non-repudiation:* This is a Bitcoin-type feature that allows users to gain access to ledgers that can be used as proof in situations in which objects or users are required not to dispute a procedure [30].

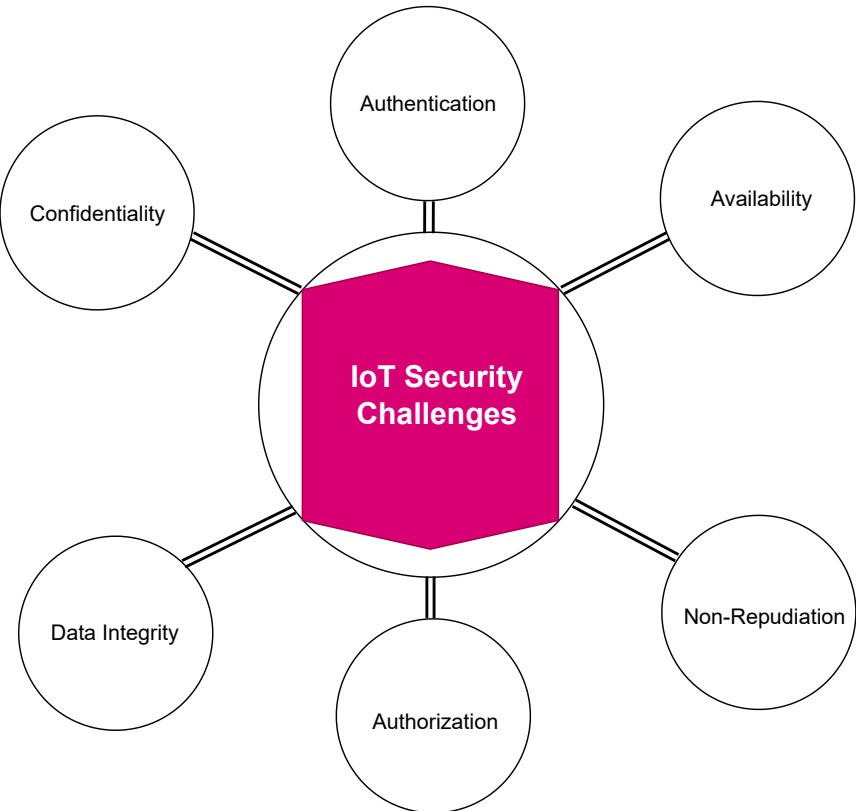

**Figure 3.** Security requirements for IoT.

### 2.2. Deep Learning Methods for IoT Security

Approaches to deep learning can be separated into three types: supervised [31], unsupervised [32], and hybrid [33] approaches. Typical supervised deep learning methods include various kinds of neural networks, such as ANNs, convolutional neural networks (CNNs) [34], and recurrent neural networks (RNNs) [35]. Typical unsupervised learning methods include autoencoders (AEs) [36], restricted Boltzmann machines (RBMs) [37], deep belief networks(DBNs) [32], etc. Hybrid methods normally combine supervised and unsupervised approaches. Popular hybrid methods include models include generative adversarial networks (GANs), ensemble deep learning (EDLN) methods, etc. [33]. As previously mentioned, by using a large number of layers and designed activation functions, deep learning is able to learn complex patterns among data, indicating that the performance of current smart applications in the IoT has been significantly improved by deep learning methods in comparison to more conventional machine learning methods [38].

Deep-learning-enabled smart applications have been studied and developed in many different areas including to secure IoT systems. For instance, by analyzing the radio frequency data of numerous radio devices, it is possible to determine their individual properties [39,40]. As an added bonus, encouraging conclusions have been drawn in the area of security vulnerability monitoring [41]. The IoT is a network that links computers, appliances, and other physical objects together [41]. Many modern IoT edge devices produce copious amounts of sensitive data in the course of their communication perception duties [42,43]. We can use the information gathered from such devices to make better decisions, boost output without sacrificing quality, and cut down on wasted energy. However, there are still many challenges in conducting effective big data analysis over the IoT data without violating users' privacy [44]. Automatically recognizing patterns in processed data and solving a number of data mining problems is made possible by deep learning techniques and algorithms [45]. Overall, as deep learning has achieved exceptional success through the analysis of IoT data streams, it is one of the most promising long-term solutions [33].

*2.3. Blockchain in IoT Security*

A blockchain is a database of ledgers that allows users to record information such as asset and transaction registries in a fully decentralized manner using a peer-to-peer (P2P) network [14]. Typically, the blocks are linked together chronically, and each block stores data for a certain period. Each data item, e.g., transaction, is written by a particular user in the blockchain, i.e., a miner, following designed protocols and can be verified by all users in the blockchain. A blockchain relies on cryptographic proof using mechanisms such as elliptic curve cryptography (ECC) and SHA-256 hashing to guarantee data authenticity and integrity [46]. The data in each block normally consist of a ledger, e.g., transaction log, and a block header, e.g., a hash code, that links to the prior block. The blockchain records every single trade ever made and facilitates global distributed trust across borders. Centralized authority and services, sometimes known as "Trusted Third Parties", are vulnerable to being hacked, compromised, or otherwise adversely affected by external interference. Even if they are trustworthy now, they may behave badly or become corrupt in the future.

The blockchain relies on a certain kind of user called a miner to create new blocks and validate and record each transaction in the block. Normally, any user on the blockchain can compete to be a miner of the blockchain by implementing certain computing tasks, e.g., solving a math puzzle. In particular, to append a new block to the blockchain, miners typically need to implement a designed task that involves finding a random number (nonce) that leads the hash of the block header to be less than a certain target. Typically, the blockchain adopts the longest chain rule, where the data stored in the longest chain is considered to be valid. This is based on the assumption that the majority of the users are honest and they will append new blocks to the block with valid data. For instance, in Bitcoin, after six confirmations or blocks, the data in the blocks are confirmed because the probability of the data being replaced is low enough. On the other hand, private blockchains have been proposed. In a private blockchain, there are permitted nodes who are trustworthy and communicate to reach a consensus. A private chain makes this process much faster, but essentially, it relies on the truthfulness of the permitted nodes.

As demonstrated by the aforementioned designed protocols, the three most notable features of a blockchain are decentralization, transparency, and immutability [47]. Due to the decentralized nature of blockchains, no single entity can claim ownership over the network. To clarify, immutability denotes that the data stored in the blockchain can never be tampered with. Decentralization means that the system never needs a central authorized party, e.g., a server. Transparency means that any user may view any other user's transactions simply by knowing their public address. These features enhance the safety and utility of the blockchain.

A common blockchain architecture is presented in Figure 4. There are two essential parts of a typical block: the block header and the block body. The former can be considered the identifier of the block, which typically contains a timestamp; a hash code of the data in the block, e.g., transactions; the hash code of the previous block's header; and a nonce value. The latter normally holds the data, e.g., a list of transactions. The data are stored in a Merkle tree structure. The Merkle root denotes the root of a hash tree with a hash of all transactions at the leaves, serving as a summary of all transactions. The identifier for a block is the hash of the block header. Additionally, the nonce value is found by miners such that the block hash can satisfy certain requirements based on the consensus protocols. For instance, in Bitcoin, every user can check that the hash of the block header is less than a target. Finding valid nonce values is normally difficult and computationally intensive. This is the so-called proof of work for miners, which reduces the possibility of miners recording invalid or bogus data in the new block. The difficulty of mining can be adjusted. In Bitcoin, the difficulty is adjusted every 2016th block so that the average of mining time for a block in the next 2016 blocks is as close to 10 min as possible.

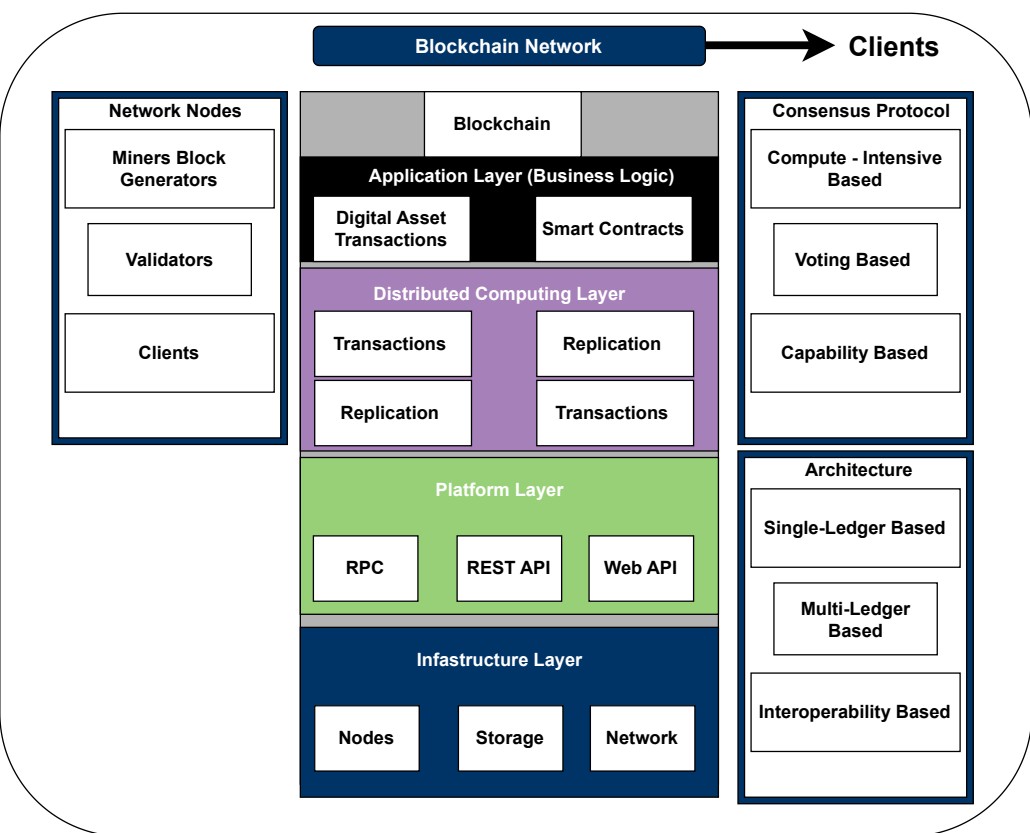

**Figure 4.** Blockchain architecture.

Bitcoin is one of the first and most widely adopted blockchain applications. Many of today's most popular cryptocurrencies use the bitcoin blockchain as their foundational platform and technology. With the introduction of the Ethereum blockchain, which includes support for smart contracts, future blockchain applications have expanded exponentially. In July of 2015, the Ethereum blockchain was released to the public. Since then, a slew of other blockchain platforms with support for smart contracts has appeared. Hyperledger [48], Eris [49], Stellar [50]], Ripple [51], and Tendermint [52] are just a few examples.

Other than Bitcoin, Ethereum's blockchain can also keep track of historical data and, more significantly, execute smart contracts. Nick Szabo is credited with being the first to use the term "smart contracts" in 1994 [53]. To put it simply, smart contracts, in a basic form, are user-created computer programs that are uploaded to and executed via the blockchain, e.g., a scripting or programming language such as JavaScript. Ethereum is serviced by

Ethereum virtual machines (EVMs), which function as the network's miner nodes. These nodes are able to reliably execute and enforce these programs or contracts in a secure manner that is immune to tampering. Ether, Ethereum's digital currency, is supported. Just as in Bitcoin, users of Ethereum can send currencies to one another by means of regular transactions that are recorded on the ledger; with bitcoin, a blockchain state is not required for such transactions.

Smart-contract blockchain-empowered applications have also been adopted in various fields, e.g., digital identification and voting, cryptocurrency exchange, system status analysis, fraud and fault detection, etc. The demand for and studies of blockchain technology-empowered applications are growing rapidly. Blockchain-empowered insurance solutions such as that provided by SafeShare [54] represent one such example. Hyperledger Fabric is also being used by IBM in the launch of its blockchain framework [48]. This platform, unlike many others, allows for the creation of blockchain apps without the need for any sort of coin. Banks, supply chains, and transportation businesses are all making commercial use of IBM's blockchain technology.

## 3. Deep-Learning-Empowered Security Solutions for IoT Systems

In his section, we introduce the implementation of deep learning to secure IoT systems in a variety of ways, including intrusion detection, malware analysis, etc.

**Table 1.** Deep learning for abnormality and intrusion detection.

| Refs. | Techniques | Dataset | Accuracy | Limitations |
|---|---|---|---|---|
| [55] | CNN | Bot-IoT | 91.27% | Wit the use of a batch size of 32 or 64, accuracy suffers. |
| [56] | FNN and SNN | Bot-IoT | 95.91% | Based on the normalization of features in the Bot-IoT dataset, we can conclude that accuracy would be less than 50% in practice. |
| [57] | FNN and SVC | Bot-IoT | 99.414% | Shown to be less effective than alternative approaches in protecting against key-logging attacks and data theft in both binary and multi-class classification, with only 88.9% accuracy achieved by the latter. |
| [58] | BiLSTM | Bot-IoT and UNSW-NB15 | 98.91% | When faced with high volumes of network traffic, IDS alarms and detection of complex attacks suffer. |
| [59] | DCNN and LSTM | N-BaIoT | 97.84% | Unable to detect emerging attacks. |
| [60] | LSTM | N_BaIoT-2018, CICIDS-2017, RPLNIDS-2017, and NSL-KDD | 99.85% | A longer training time is required for large datasets. |
| [61] | DNN | 4 IoT datasets | 99.75% | Attack test limited to scanning, DoS, MITM, and Mirai). |
| [62] | LSTM, RF | Smart Fall dataset | LSTM: 93.4%; RF: 99.9% | When compared to other approaches, LSTM is widely regarded as having subpar accuracy. |

### 3.1. Abnormality and Intrusion Detection

Several machine-learning-based strategies are still being used today to detect outliers, anomalies, and intrusions in various IoT platforms [56]. For instance, traffic filtering is a popular method to detect intrusions, which separates legitimate packets from malicious packets using per-packet or batch analysis. Traffic classification is effective, but the process generates more false positives than is typical, making the method less reliable than it would otherwise be. Behavior-based models, on the other hand, are used in the process of locating network intrusions. Both algorithms are adopted in the context of the IoT. For example,

Meng [63] presented a trust-based model to detect intrusions in IoT systems. In contrast to classic methods, the developed approach considers not only the nature and category of traffic but also the level of trust that one has in a given device. As a method for detecting intrusions in IoT networks, combining trust management and traffic classification was suggested in the paper.

In light of the fact that traditional methods have been shown to be ineffective against adversarial attacks, the authors of [56] implemented a type of deep learning called a self-normalizing neural network (SNN) and a feed-forward neural network (FNN) to detect intrusions. A new method of network intrusion detection and traffic analysis using an FNN was proposed in [55]. Using the Bot-IoT dataset, the authors compared the performance difference between FNN and SVM models. The results showed the proposed solution was less effective at preventing key-logging attacks and theft of sensitive data via binary classification.

As previously mentioned, deep learning has become an important tool in intrusion detection systems (IDS), especially when it comes to heterogeneous IoT networks. The typical structure of deep-learning-based IDS is shown in Figure 5. For instance, Kim et al. [64] applied long short-term memory (LSTM), a type of Recurrent neural network (RNN), to detect intrusions. In a similar study, Saeed et al. [65] considered the low computing powers of IoT networks and developed a neural network in a two-tiered structure that can efficiently detect anomalies without consuming a large amount of power. In particular, the first layer in their neural network aims to teach the system normal operation, and the second layer looks out for illegal memory access (IMA). Susilo et al. [55] used deep learning methods such as CNNs. Zhao et al. [66] suggested intrusion detection by combining a deep belief network (DBN) with a probabilistic neural network (PNN). These studies, to some extent, do not adequately cover the entire spectrum of attacks, including zero-day exploits.

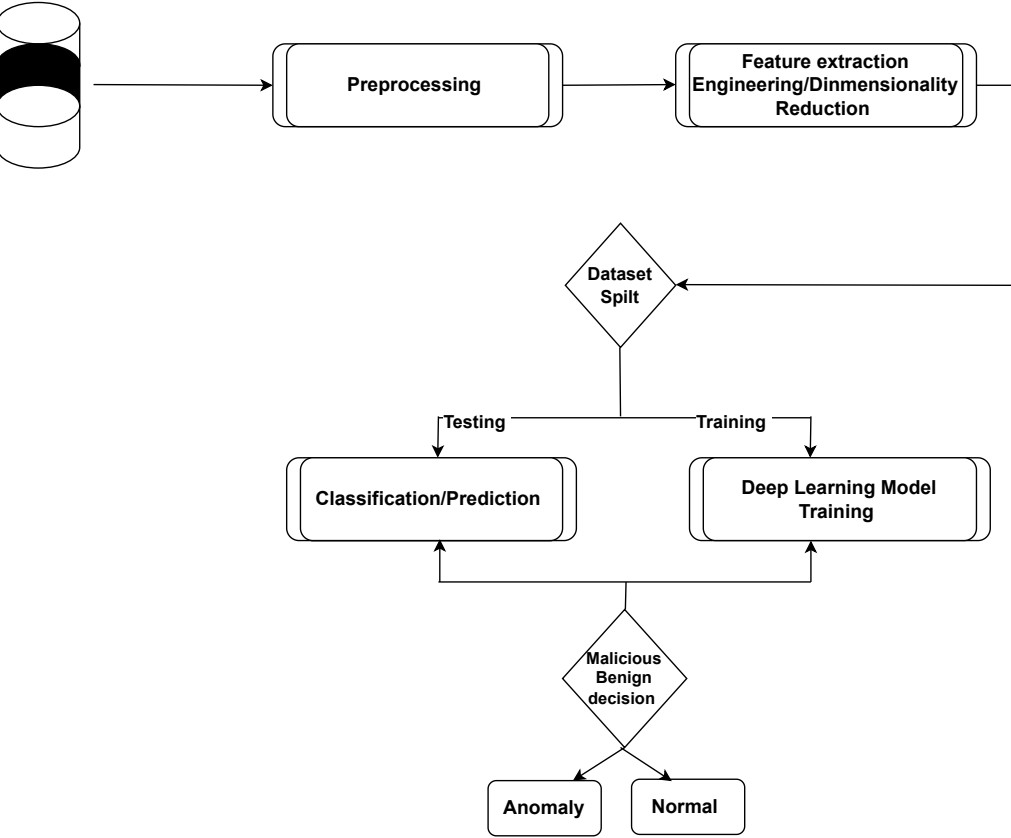

**Figure 5.** IDS based on deep learning.

With an increasing number of devices hooked up to the IoT, the danger of zero-day attacks is introduced. In this context, Samy et al. [60] proposed the use of LSTM for detection of unknown network attacks. They examined this method alongside four other deep learning models and five IoT datasets: GRU, LSTM, CNN, CNN-LSTM, and DNN. The proposed model requires an extremely large set of training data and considerable time for training. Network traffic over IoT networks presents a risk, as discussed by the authors of [67]. Several machine learning techniques were adopted for the testing in this work, such as decision trees, naive Bayes, and deep learning.

On the other hand, the distributed and decentralized structure is a new challenge for traditional security approaches. As stated in [58], typical IoT security approaches mainly detect attacks on centralized systems, and the algorithm is normally launched on either edge devices or a central server, which may limit the attack detection success rate, especially detection of attacks launched in a distributed manner, e.g., distributed DoS in IoT. To address this issue, the authors of [57] developed distributed convolutional neural networks (DCNNs) for smart edge devices and deployed the LSTM model on the central server to identify distributed attacks launched from either side.

Although deep learning has shown its strong ability, a major drawback of using a deep learning approach is its high reliance on the sufficiency and quality of the data [59]. This was also demonstrated in [68], where the authors trained both a deep learning model, i.e., LSTM, and machine learning algorithms, i.e., random forest and AdaBoost. The results showed that by using a single dataset with a limited amount of data, the accuracy of LSTM was lower than that of other machine learning models. However, this issue can be mitigated by using a larger set of data and more informative data. On the other hand, some works considered employing ensemble learning to help enhance the accuracy of deep learning models. We provide a brief summary of the main studies included in this section in Table 1.

### 3.2. Detecting Threats and Taking Preventative Action

It is possible to launch multiple layers of attacks against IoT systems, one of which is the hacking of discreet devices. For instance, major attacks, e.g., ransom attacks, can be launched on one layer, and further attacks, e.g., sophisticated attacks, can be launched on additional layers. The cryptographic primitives used by conventional attack mitigation and detection methods are imperfect and prone to false positives. Therefore, many methods have been attempted, e.g., K-NN, SVM, ANN, etc. Taking advantage of foggy conditions, Diro and Chilamkurti proposed a deep-learning-based attack detection algorithm for the IoT in [69]. The attack detection method is essentially launched in smart edge devices in a distributed manner. The proposed method aims to implement deep learning models as close to the network edge as much as possible such that the model can provide the output quickly and accurately. Chilamkurti et al. [69] similarly proposed another distributed deep-learning-based attack detection mechanism for the IoT. Based on fog computing architecture, they put into place deep learning methods for attack detection. The proposed method places an emphasis on dialogue between the fog and the objects. For optimal resource utilization and minimal communication latency, a detection mechanism should be deployed in the fog layer, where the learning module is also activated. Hafeez et al. [70] developed an anomaly identification method based on the combination of the clustering method and RNN models. With the help of deep learning, the method suggested in [71] can spot DGAs. In particular, a series of models, e.g., LSTM, GRU, and CNN, was applied in this study. For the purpose of multiparty computation and secure IoT devices, the research effort reported in [72] presented a federated architecture that operates using a deep learning technique. We provide a brief summary of the main studies included in this section in Table 2.

**Table 2.** Deep learning for detecting threats in the IoT.

| Ref. | Algorithm | Results | Limitations |
|------|-----------|---------|-------------|
| [69] | Security for the IoT based on deep learning, with support for both deep model (DM) and shallow model (SM) attack detection | Accuracy: 99.20% Precision: 95.22% | The dataset used in this work is representative of legacy network architectures devoid of IoT traffic and related attacks. |
| [73] | IoT malware detection using a deep-learning-based long-short-term memory (LSTM) algorithm trained on Opcodes. | Accuracy: 98% | The dataset used in this work was extremely small. |
| [61] | Recurrent neural networks for intrusion detection using deep learning (RNN-IDS). | Accuracy: 97.09% Precision: 83.28% | A long testing time is required for detection. |
| [74] | Federated DL-based method for multiparty computation that takes into account the safety of IoT devices. | Accuracy: 56% | The detection accuracy achieved by the algorithm is unsatisfactory. |
| [75] | D'IoT, which is a proposed autonomous self-learning system for identifying vulnerable IoT devices. | Accuracy: 95.6% Precision: 92.10% | The reliability of detection is poor. To improve detection accuracy, a method of feature selection is required. |
| [73] | A recurrent neural-network-based anomaly detection system. | Accuracy: 98% | It is not possible to factor in testing time. |
| [76] | A new technique for packet-level detection in the IoT and networks was developed using deep learning and bidirectional long short-term memory (LSTM). | Accuracy: 99% Precision: 98% | Testing time was not calculated. |
| [77] | Method for mobile malware detection using Q learning for efficient offloading. | Accuracy: 67% | Predictability is too low. |
| [78] | Model based on deep learning that uses LSTM to detect bots by leveraging content and metadata. | Accuracy: 90% | The reliability of detection is poor. |
| [66] | Technique for detecting intrusions using deep learning based on the combination of a deep belief network and a probabilistic neural network. | Accuracy: 99.14% | The values of FPR, FNR, FDR, and FOR are missing. |

Regarding the identification of a compromised IoT, Nguyen et al. [75] presented a self-learning system called D'IoT, which was tested on a self-generated dataset. HaddadPajouet et al. [73] proposed the use of LSTM to search for Opcode-based IoT malware. To test the efficacy of the proposed framework, Opcodes for running IoT applications built on the ARM architecture were used. This approach uses text mining, a feature selection technique, to extract a useful feature vector from Opcode. In addition, Diro et al. [69] incorporated both deep and shallow learning models to identify an intrusion in the IoT. Pektacs et al. [62] identified a botnet by conducting an analysis on the statistic data of the communications among nodes in the IoT, e.g., packet info, transmission duration, etc. Using both content and metadata, Kudugunta et al. [78] presented an LSTM-based method to identify bots.

Attacks on IoT devices can be identified by patterns in their operating codes (Opcodes), as demonstrated by Azmoodeh et al. [79]. To distinguish between safe and harmful programs, they first converted the Opcodes into a vector space and input the vector to the deep learning model. In their study, a CNN with an Adaboost classifier was utilized as the algorithm. In addition, McDermott et al. [76] adopted packet information and employed a bidirectional LSTM to assess whether the packet was harmful. In particular, they used traffic information from both the botnet and regular IoT in their experiment. Xiao et al. [77], on the other hand, develop a Q-learning-based malware detection method for mobile devices that does not require any prior knowledge of the mobile device's trace generation and bandwidth. The author investigated the static malware detection game and its Nash equilibrium (NE) on a time-variant wireless network.

### 3.3. Preventing Denial of Service (DoS) and Distributed Denial of Service (DDoS) Attacks

It is believed that the IoT is a "Land of Opportunities for DDoS Attackers" [80]. The number of attacks on IoT infrastructure has dramatically increased on a never-before-seen scale. Mirai, one of these attacks, nearly brought the Internet to a halt by using common household items as bots to carry out DDoS attacks against a number of different corporations. Babycams, printers, and webcams were among the affected items. Other bots that resemble Mirai in [9]. Because of the sophisticated method that was used to spread throughout the IoT network, the Mirai malware family caused significant damage to Internet services. On the other hand, due to the wide variety of designs, developing a unified method to defend against DDoS attacks in the numerous IoT platforms is extremely difficult.

Significant research accomplishments have been produced through the application of a variety of strategies to resist DDoS attacks in the IoT. For instance, Shaaban et al. [81] proposed a deep-learning-based DDoS attack detection approach using LSTM and GRU. Compared to conventional ML models, their proposed system achieves superior results. Using deep learning methods such as a stacked autoencoder and stacked restricted Boltzmann machine (RBM), Nugraha et al. [82] created an NIDS that detects intrusions based on anomalies in network traffic. Van et al. [83] proposed a novel framework for detecting botnets on TCP/UDP/IP packet flows that makes use of deep neural networks (DNNs) and ladder networks. It presents superior results when compared to traditional methods of detecting botnets. Hodo et al. [84] similarly adopted deep learning to mitigate both DoS and DDoS attacked in the IoT. Likewise, Meng [63] employed a multilayer perception (MLP) approach to detect DoS attacks in a common IoT platform, e.g., a sensor network. Both shallow and deep learning methods were investigated in [64]. The authors used probability estimates based on shallow learning methods to foretell intrusions. They tested a convolutional LSTM deep neural network and found that it performed significantly better than the shallow learning techniques they had previously used.

Priyadarshini et al. [85] developed a source-based method to defend against DDoS attacks in fog computing. Their proposed work employs LSTM to identify legitimate and malicious packets by analyzing the patterns in the time series data. Similarly, Sabeel et al. [86] combined DNN and LSTM to detect unknown DoS and DDoS attacks. Doriguzzi et al. [87], with the aim of reducing computation consumption, developed a lightweight and usable CNN for DDoS detection called LUCID. CNN properties were used to detect whether traffic flows were attacked or safe. Their results demonstrate the viability of their proposed method for DDoS attack detection in operational settings with limited resources.

### 3.4. IoT Access Control and Authentication

Authentication is another crucial requirement for IoT security. Normally, before users can access IoT applications and/or services, their actions have to be validated. Access control is critical in both IoT and traditional networks. However, because of various factors, e.g., network heterogeneity, network volume, device resource limitations, network (IoT) security, etc., implementing access control can be difficult. Furthermore, the ability to grant and suspend a user's access to specific critical data contained within IoT apps and services is critical. For example, Shi et al. [88] attempted to utilize the information of human identity, as well as the recognition of human activities based on the channel state information from WiFi signals, and developed a human physiological activity-based user authentication method. In addition, the authors developed a three-layer DNN-based authentication method that also takes human physiological activities as the input.

It should be noted that factors such as the sampling rate and training size considerably influence the accuracy of authentication in DNN-based methods. For example, Das et al. [89] attempted to use LSTM for authentication in power-constrained IoT networks. LSTM is used to analyze the flaws and vulnerabilities of hardware that affect signal strength. The designed deep learning model is capable of identifying users by detecting and exploiting these flaws. It is critical to assess such a solution in light of the adversary's

challenges. In addition, Chauhan et al. [69] presented an authentication method based on an RNN for use in smart home applications in a similar manner. This method employs both acoustics and voice commands. As their primary method of acoustic authentication, the authors concentrated on developing a simple RNN system. The authors also compared and contrasted the SVM-based approach and the LSTM approach for user authentication in a smart home environment. These works explain why deep learning algorithms outperform more commonly used ML algorithms in authentication tasks. Regarding accuracy, the LSTM approach generally outperforms DNNs and regular RNNs in experimental scenarios. Nonetheless, contrary to appearances, this contrast is only valid in the context of such settings. Furthermore, each deep learning approach has its own set of drawbacks and shortcomings that may affect how well it performs. These include, for example, the DNN's training size and the LSTM's line of sight.

### 3.5. Dynamic Language-Based Malware Analysis in IoT

Another important field of application of DL for IoT is malware analysis. Haddad et al. [73] proposed an RNN-based method for analysis of malware in the IoT. They considered applications for the IoT based on advanced RISC machines (ARM). In a similar manner, Azmoodeh et al. [79] employed a deep convolutional network to investigate the operational code (Opcode) sequence of an IoT subtype known as the Internet of Battlefield Things (IoBT). In addition, Karbab et al. [90] developed a deep-learning-based malware analysis framework for Android applications called MalDozer. Malicious and trustworthy applications for the Android platform were used to validate the detection framework, which is based on ANN. MalDozer's fundamental building blocks are method calls to the Android API and requests for permission to access resources. Su et al. [91], on the other hand, proposed using image recognition as a DDoS virus detection method for IoT networks. The creators of this method begin by collecting and categorizing two major families of malware: Linux and Mirai. They attempted to convert the binary files of the IoT applications into grayscale images, then applied the CNN on which to detect whether the photographs were from malicious or benign software.

## 4. Decentralization in IoT Security

So far, we have reviewed the literature on implementing deep learning to address variant IoT vulnerabilities. However, we realize that most conventional deep learning methods are implemented on a centralized structure, such as a system containing a central server [39], which presents a challenge due to the high communication overhead between devices and the server. To enable deep-learning-based tools, data from geographically dispersed IoTs is typically transferred from the asset and stored on a centralized computing platform, which places a greater strain on communication networks, leaves more data vulnerable to attacks, and compromises user privacy. To solve this problem, we need to move away from centralized deep learning and toward a more distributed system that can process sensing data close to the source, at the edge, and on IoT devices.

This transition is met with two major obstacles. First, many IoT applications are subject to concerns and potential vulnerabilities regarding security and privacy. As a result, not only accuracy but also speed, security, and privacy preservation are essential performance metrics for decentralized deep learning. Second, the data collected by a variety of edge devices or sensors may be heterogeneous, which means the data for a certain learning task could be biased and inconsistent across devices [92]. This is common in practice because such data are usually not evenly distributed or are non-independently and identically distributed (non-IID). Certain sensors may only collect or focus on parts of global features and hence may be only adept at parts of the learning task. Therefore, decentralized deep learning methods must be resilient to non-IID data samples while learning the task.

### 4.1. Blockchain Solutions and IoT Security

A blockchain and smart-contract-based IoT system was developed by Pan et al. [93]. The created architecture makes use of blockchain technology and a cryptocurrency system to replace the regular cloud server in the IoT. IoT-related transactions and activities are kept in blocks so that every user in the system is able to verify them. On the other hand, smart contracts are implemented for actions in accordance with predetermined policies and roles. The examination of the framework proved the extensive safety of IoT applications. Latif et al. [94] proposed a blockchain-empowered IoT network architecture that is strong, lightweight, and distributed. The architecture is reliable for various activities, e.g., transaction recording, user authentication, access control, etc. Fan et al. [93] presented a system in which IoT devices can verify each other's identities and exchange data securely thanks to blockchain technology. However, not all authentication methods for enhancing the security of IoT data transfer were considered in the study.

In [95], Huang et al. presented an authentication system that makes use of blockchain technology and networked identities. Blockchain anonymity was used to assess the system's block speed and performance time. Using a connection protocol, Zhang et al. [96] proposed a blockchain system. The proposed protocol proved effective in ensuring the security of IoT network endpoints. It also included a blockchain-based network protocol to provide cryptographically secured authentication and validation of inter-IoT terminals and access to platforms. According to Qian et al. [97], integrating AI with blockchain helped further the objective of safe and scalable IoT systems. For large-scale data analytics tasks in IoT applications, the proposed architecture provides precise, centralized, secure, private, and low-latency options. Latency and insufficient computing power are addressed to some extent but not fully. Agrawal et al. [98] presented a blockchain framework that aims to continuously ensure the security of IoT systems without human intervention. Gong et al. [99] studied different types of IoT computing devices using a fundamental four-layer IoT blockchain modeling strategy, which includes perceptual, network, blockchain, and application layers. As a proof of concept, they proposed testing a fully automated trading device built on the Ethereum blockchain. As demonstrated by the developed model, blockchain has enhanced the confidentiality, verifiability, and safety of IoT applications.

Considering privacy preservation in IoT, Lin et al. [100] designed a blockchain-based thin-client system to address the IoT's authentication and privacy issues. Because of the suggested method, the issue of thin-client devices' limited storage space can be remedied. The trustworthiness of the plan was also verified by means of a comprehensive security examination. However, further study is needed to completely grasp how smartphones are impacted by the computational load of thin clients. Another blockchain technology was introduced by Rathee et al., in [101], to ensure security and privacy. A wide range of IoT systems can now be resistant to attacks because of the proposed blockchain-based methods. Blockchain technology further makes it easier to implement IoT activities across different domains and layers, e.g., from the fundamental (data collection) layer to the higher (communication and applications) layers.

There are also many works focusing on using blockchain for the security of e-business or financial IoT. For instance, Bernabe et al. [102] presented a blockchain-empowered system for safeguarding financial and trading transactions in the IoT. The presented method builds a peer neighborhood network to make the blockchain ledger easily fetched by all users. After putting the suggested method through its paces, the universal peers, the size of ledgers, and the weight of blocks can be effectively decreased. By more evenly spreading work, it accelerated P2P financial dealings. Huang et al. [103] utilized a credit-based blockchain platform for industrial IoT. The method makes use of a credit-based PoW protocol, guaranteeing the system's reliability and the swiftness of transaction processing. The Raspberry Pi framework was introduced, and a case study of an intelligent factory was carried out using this mechanism. Likewise, Qian et al. [97] designed a blockchain-empowered framework for securely managing interconnected devices. The suggested approach makes it simple to assess applications, networks, and user perceptions independently, which is

important because security problems might arise in any of these areas. Zhang et al. [104] presented an IoT e-business concept. The model looked at the stages typically involved in an e-business transaction, including preparation, agreement, and performance. When asked how to make IoT e-business more widely available, the authors proposed using a peer-to-peer transaction mechanism such as blockchain. Furthermore, they created a smart-contract-based approach to facilitate the exchange of confidential financial data and intellectual property between IoT e-businesses.

### 4.2. Applications of Blockchain in IoT

In this section, we introduce various applications for which Blockchain has been deployed for IoT devices, as shown in Figure 6.

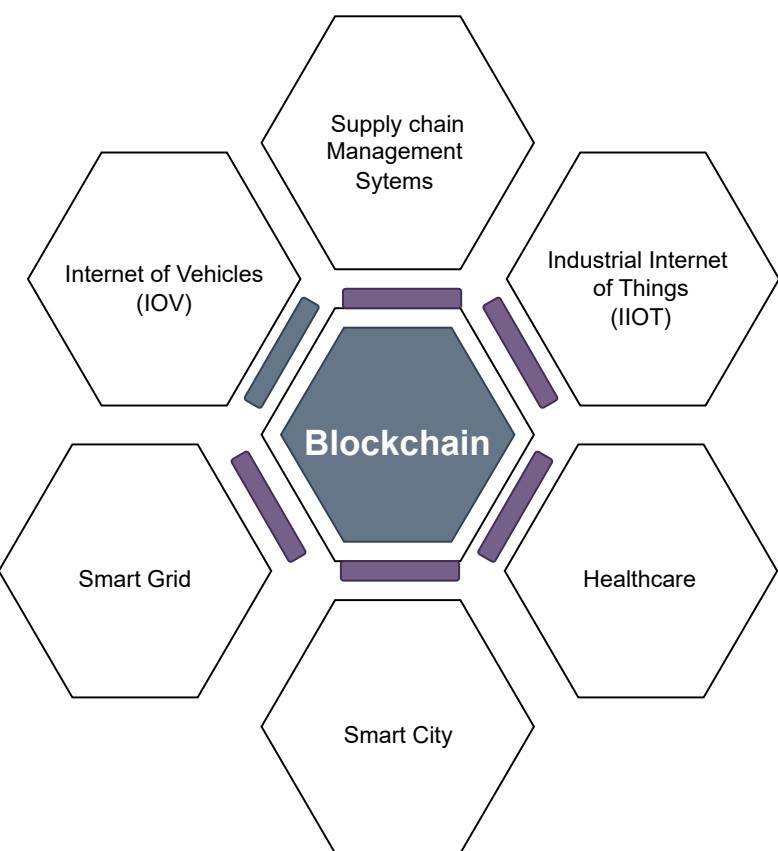

**Figure 6.** Application of blockchain in IoT.

*Internet of Vehicles (IoV):* Many of the security and trust difficulties with Internet-connected vehicles could be resolved by implementing blockchain technology. Basic trust and security characteristics, such as vehicle identity management, reputation, communication channel integrity, system automation, etc., need to be guaranteed by a successful, dependable system [105,106]. Social IoV (SIoV) is one such common example [107]. Smart contracts are adopted in this concept to implement tasks such as managing access, monitoring activities, etc. As an added bonus, privacy management via smart contracts is possible [108].

*Managing Supply Chain:* Blockchain-empowered IoT devices can help improve the efficiency of the supply chain over the whole process, i.e., from the factory to the end user. IoT has shown strong potential to improve supply chain systems. However, there is an urgent demand to ensure the integrity and trustworthyness of data in such systems. According to to [109] and other sources, blockchain technology offers a definitive answer to this problem [110]. Particularly, smart contracts allow us to effectively build a fully automated supply chain management system without relying on any kind of centralized authority.

*Industrial Internet of Things (IIoT):* Integrating smart edge devices into industries enables monitoring and maintenance of all processes with no need for human intervention [111]; this is where blockchain comes in. The integration of blockchain can further improve the reliability of the IIoT in terms of data trustworthiness, access control, privacy preservation, and security [112,113].

*Firmware Updates in IoT:* The updating process of devices in IoT is also vulnerable to disruption, and the flow of code execution can be altered. There are also difficulties in distributing accurate firmware updates due to the restricted capabilities of IoT devices [6]. Another challenge is scalability; it is not feasible to manually update hundreds of endpoints. The majority of firmware update procedures rely on asymmetric cryptography technologies, which are notoriously difficult and resource-intensive. Regarding its decentralized structure, blockchain can effectively eliminate single points of failure [114,115].

*Smart Healthcare:* A significant part of modern medical practice involves implanting sensors and other devices into patients for monitoring and diagnostic purposes. The IoT enables integrated sensors to gather data from patients to be shared with the doctor, who can then evaluate the patient's condition remotely. By taking this route, the patient can be freed from the hospital's hierarchical structure while still maintaining close contact with his or her physician. Strong interest exists in implementing such a cutting-edge IoT framework to guarantee real-time monitoring of patients, given the aging population and rising medical costs [11]. The Food and Drug Administration (FDA) and IBM Watson Health have collaborated on the use of blockchain technology to secure oncology-related data. They found that blockchain is able to store data collected from various sensors when all necessary conditions are met [116]. The healthcare industry is likely to adopt blockchain technology for safe storage of patient medical records. Blockchain can offer patients with unchangeable record confirmation and confidence in such a way that sensitive patient information can only be accessed by authorized personnel following designed protocols.

*Smart Grid:* Recent increases in electricity production can be attributed to the utilization of cutting-edge IT methods that take into account consumer needs via the power grid. This distribution network, also known as a "smart grid", was developed to both increase the reliability of service provided to final consumers and to reduce the wasteful use of energy during generation [117]. It is a system in which power plants and consumers are linked via a network to ensure that supply and demand are always in step [11,117]. Similar to other IoT systems, security and privacy are also major concerns in a smart grid due to its inherent complexity and dissemination of sensitive information [4,118]. As a result, setting up a secure, non-centralized, distributed infrastructure is essential [119]. Blockchain is able to benefit the smart grid in many aspects, e.g., both security and privacy guarantees, improved scalability, efficiency, etc. Transactions are securely stored in the blockchain due to its consensus protocol. Given the difficulty of altering or removing a block in the blockchain, the system is reliable and secure once an electricity transaction has been recorded. There is much less exposure to attacks than with a conventional, centralized data architecture because there is no single point of failure. Electricity producers and consumers alike can benefit from the increased clarity in pricing resulting from the widespread adoption of blockchain-based trading infrastructure.

*Smart City:* The term "smart city" describes a community in which digital tools are used for problem solving and management [120,121]. By using IoT, a smart city can provide better service quality with lower computational overhead. In the meantime, blockchain has been studied to enable decentralized smart city applications and integrate with cutting-edge urban technology. A connected intelligent city network generates tremendous data through IoT devices, which are then analyzed in distributed blockchain databases. Typical privacy safeguards are not easily implemented in smart cities. One of the most practical solutions in a smart city is lightweight privacy security that provides strong secrecy protection and data utility power. In blockchain-based smart city initiatives, smart contracts are viewed as a potential technological advancement [11].

### 4.3. Blockchain Solutions for IoT Trust Issues

*Authentication:* To guarantee that no two nodes are fraudulently exchanging data with one another, authentication is performed at both ends of the connection. When choosing an authentication method in the IoT, it is important to take into account the system's hardware, connectivity, power consumption, and security requirements. Because of the preinstalled nature of the keys, the symmetric key model is well-suited for low-resource, low-security devices. When data storage on a device based on hardware needs to be protected, hardware security modules (HSMs) are among the safest options. A trusted platform module (TPM) is a widely used way to store asymmetric keys in unhackable hardware. However, the models are not infallible because they can be tampered with by malicious users. Blockchain, a distributed ledger technology, provides a practical alternative because its data are immutable. As an internal network, a private blockchain can protect the IoT from outside attacks. Successful blockchain authentication in the IoT was implemented using the Authenticated Devices Configuration Protocol (ADCP) defined in [122].

*Identity Management:* For identification purposes, an identity produced by an identity management system must be able to pass an external authentication process. Common identity management architectures can be broken down into three types: standalone, federated, and centralized [123]. There are three main sorts of people involved in identity management systems: the identity owner, the identity issuer, and the identity verifier. Security flaws in traditional identity management systems such as those described above arise because of the centralized manner of storing information [124]. With blockchain technology, several problems with present identity management systems can be solved [123]. Blockchain-based identity management solutions rely on decentralized identifiers (DIDs) for the authentication of users' online personas [124]. DIDs are digital identifiers that are appended to credentials by an identity issuer and stored alongside those credentials in a blockchain-based system [125]. In this scenario, blockchain supports universal access to a shared ledger of records while simultaneously maintaining strict controls over who can view or modify individual records. There are some existing identity management platforms, e.g., uPort [126], Sovrin [127], ShoCard [128], etc., that allow users to create their own identities on the Ethereum blockchain.

*Data Integrity:* There are a few widely used cryptographic protocols, e.g., the Secure Hash Algorithm (SHA), Rivest–Shamir–Adleman (RSA), etc. However, IoT devices might not have sufficient CPU capacity to perform such complex calculations. Additionally, because IoT nodes are not always running, adversaries have a window of time to compromise information. The authentication and read–write safeguards proposed as a solution may not be practical for all IoT applications. While software-based solutions such as multilevel security (MLS) and hardware-based solutions such as TPM are possible, cryptographic solutions such as TPM are also practical. However, these techniques may be subject to limitations in various complex IoT systems. Although protocols such as that proposed in [129] are used to protect the security of IoT systems, they were developed to fend off specific types of threats. Given the immutability, distributed ledger, and decentralization features of blockchain, the integrity of data in the IoT can be effectively guaranteed. In particular, once the information is registered in the block and verified by the users, it is nearly impossible for malicious users to tamper with that data [130]. On the other hand, the IoT is an exciting new frontier for the integration of private blockchain distributed ledgers to guarantee the veracity of collected data. Data from the IoT can be encrypted using smart blockchain contracts [131].

*Authorization and Access Control:* Access control must be highly reliable in the IoT because it involves ongoing queries and streaming data. The majority of methods for authorizing and regulating access rely on a mechanism that makes access choices based solely on information stored locally on the device. While device-determined policies offer more leeway in authorization models, trusting them might be challenging because of their reliance on local data. Integrating blockchain with current authorization methods provides a promising solution, which enables a solid platform for trustworthy interactions between

users and autonomous systems. Models such as those shown in [132] center primarily on issues and solutions related to authorization and access control [133]. Using OAuth 2.0 and blockchain, these approaches offer enhanced security and transparency. We provide a brief summary of the IoT security challenges and blockchain solutions in Table 3.

**Table 3.** Blockchain solutions to IoT security challenges.

| IoT Security Challenge | Solutions via Blockchain |
|---|---|
| Interoperability | Since blockchain operates in a decentralized and automated fashion, it is the foundation upon which interoperability rests. |
| Data integrity | Each node in the blockchain shares the same information and can validate it by referencing prior records. |
| Authentication | Blockchain employs asymmetric cryptography in a decentralized fashion, with each entity in the system assigned a unique hash ID that is shared publicly among all nodes, fostering confidence among the network's nodes. |
| Authorization and access control | Ethereum blockchain-based smart contracts. |
| Identity management | There are many applications for the immutability and distributed ledger technology that blockchain provides. |

*4.4. Blockchain Security Flaws*

In this section, we present the security challenges associated with applying blockchain technology; the major challenges are shown in Figure 7.

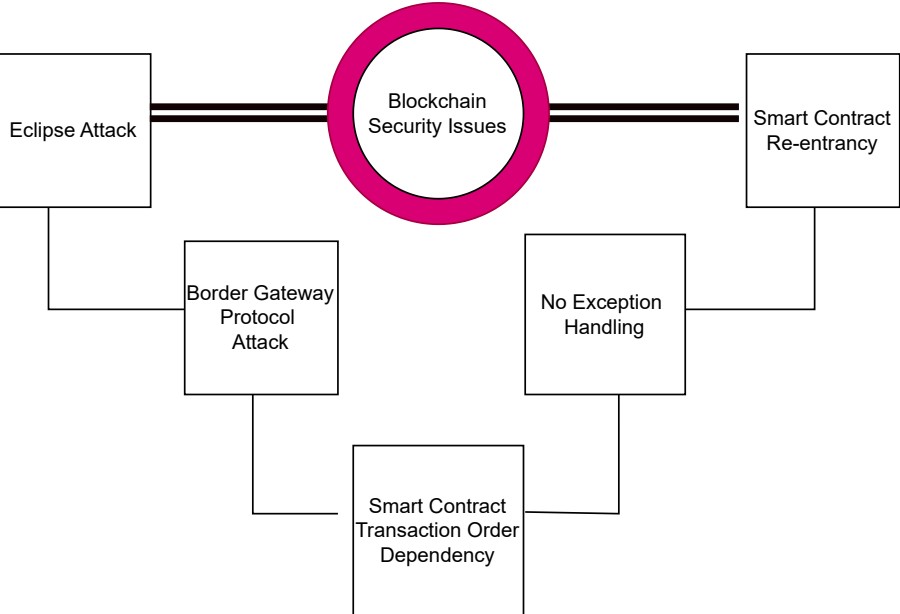

**Figure 7.** Blockchain security issues.

*Order Dependency in Smart Contracts:* In most cases, miners do not accept the order in which transactions come and instead select a new order, leading to an issue with transaction order for smart contracts that rely on the current values of storage variables [134].

*Eclipse Attack:* The victim's incoming and outgoing connections are manipulated in an eclipse attack, which distorts the true nature of the blockchain. Some forms of assault, such as botnets, can be used to carry out such an attack [135].

*Border Gateway Protocol Attack:* The blockchain network's routing information can be compromised using an attack known as a border gateway protocol attack [136]. The goal of this assault is to slow down the spread of data across a network. The other miners are left behind because of the information vacuum that opens up.

*Handling Exception in Solidity:* Many smart contracts are implemented in a programming language called "Solidity". It is possible to exploit flaws in this language due to improper exception handling for certain actions. A Solidity "SEND" operation, for instance, does not contain an exception handler in the event of a failure. Only Boolean values are transmitted, which, for some purposes, may be adequate. Texas, the place of origin of authorization phishing attacks, involves pretending to be the legitimate originator of a series of transactions in order to steal sensitive information.

## 5. Discussion and Future Challenges

In this section, we discuss some open and challenging issues that must be considered for further study.

### 5.1. Importance and Challenges of Integrating Blockchain Technology and Deep Learning

The emergence of blockchain allows deep learning applications to be applied in a wider range of fields because of the security and decentralization features provided by the blockchain. The blockchain can be considered a distributed, verified global database accessible by all nodes on the network. If machine learning models were built and deployed using blockchain technology, their development and deployment could be monitored in real time. When applied to deep learning algorithms, blockchain technology eliminates the effectiveness of data manipulation, model alterations, and other external attacks [137]. The main advantages of integrating deep learning and blockchain is summarized in Table 4.

**Table 4.** Advantages of deep learning and blockchain integration.

| Merits | Type of Blockchain | Description |
|---|---|---|
| Improved robustness | Private/public | The combination of blockchain technology and deep learning can be useful in business settings, where parties can work together in a trustless and automated manner. |
| Automatic decision making | Private | With the help of the decision traceability feature of deep learning models, verifying that choice is a breeze. Furthermore, it ensures that the documents were not tampered with during the auditing process with human involvement. |
| Joint decisions | Private/public | By employing swarm robotics to blockchain technology, robots can obtain access to a voting-based approach that can help them make an informed decision based on the data they have collected. |
| Information assurance | Private | When fed consistent data from the blockchain, deep learning algorithms can make better, more informed decisions. |

It is possible that consumers could be compensated for providing data required to train machine learning models thanks to the security mechanism provided by blockchain technology. Expert systems equipped with domain-specific knowledge bases make up the lion's share of current AI efforts. However, researchers are honing effective machine learning algorithms for application to real-world problems. When compared to deep learning, which places an emphasis on data usage to train models and provide accurate predictions, the blockchain stands out as a trustworthy, fault-tolerant technology. The immutability of the blockchain also protects deep learning models or data from cyber attacks and addresses concerns about data noise. In this study, we found that by integrating blockchain technology with deep learning models, accurate predictions can be achieved in a risk-free and trustworthy manner. Applications that are powered by deep learning can be used to collect, analyze, and use mission-critical data with the help of a secure, immutable, and distributed backbone provided by deep learning and blockchain.

However, it should be noted that there are still many challenges associated with integrating blockchain and deep learning in current IoT systems. Without solving the fundamental defects of blockchain, e.g., transaction efficiency, block size, power consump-

tion, etc., many deep learning solutions are still in their nascent stages, representing only proof of concept [138]. One of the fundamental concerns is that IoT devices might be overwhelmed by running both blockchain and deep learning, both of which intensively consume computation resources. Furthermore, most IoT applications require real-time application, which exacerbates the case of applying both blockchain and deep learning. Privacy adds another level of difficulty regarding both the data and deep learning models in the IoT.

### 5.2. Challenges of Integrating Blockchain and IoT

With the incorporation of blockchain technology, there are a few security concerns that should be noted. One of the main obstacles for businesses and developers is the dissimilarity in the technologies' underlying functioning principle and architecture (Figure 8). Thus, it is essential to first identify the problems that could raise during the integration of these two technologies.

*Size of the Blockchain:* Blockchain usually has vastly different operational and static data sizes. For example, Bitcoin and Ethereum have grown to be between 250 GB and 1 TB in size, respectively, making them prohibitively large in comparison to IoT systems. Due to the limitation of computation resources and storage of current IoT devices, it is challenging to integrate blockchain with IoT. Given its large file size, the widespread implementation of blockchain in IoT infrastructure is limited. One possible solutions is to move the data to cloud storage. However, cloud computing usually has a centralized structure, whereas blockchain is decentralized, which might lead to framework conflicts.

*Required Computing Power:* Blockchain employs PoW and consensus algorithms to provide high-security features such as immutability, robust authentication, etc., consuming large amounts of computational resources and energy to execute these algorithms. However, IoT devices employ lightweight protocols and processes that require little in the way of power. Blockchain and IoT systems use two different protocols, and they could not be more dissimilar in terms of the amount of computing power and energy required. Making them work together is a difficult undertaking in and of itself.

*Security:* Integrating blockchain technology with IoT systems has been hailed by several experts as a means of ensuring the safety of connected devices. However, the data dependability of IoT devices becomes a major challenge as a result of this integration. A typical blockchain only ensures the integrity of data in the absence of attacks. The failure of hardware, the use of counterfeit hardware, the use of compromised networks and hardware, etc., are all potential causes of data corruption in the IoT. Thus, it is important and challenging to design security algorithms for the incorporation of blockchain into IoT infrastructure.

*Challenges of Identity Management:* The management of one's own identity is essential. Weak data correlation, insufficient data, and outdated or erroneous data are just some of the information collection issues related to identification records. Items such as automobiles, houses, paintings, and digital publications all benefit from possession management for the purposes of copyright management, property rights management, and traceability. The blockchain can be used to record ownership in a way that is immutable. The blockchain method ensures effective contract execution and the subsequent tracking of property ownership. The blockchain method verifies ownership claims and establishes the existence, validity, and uniqueness of choices by using hashing algorithms and timestamps to provide immutable digital evidence such as video, text, and audio. Proper ownership must be confirmed when it has been established.

*Users' Anonymity and Data Privacy:* The potential application of blockchain technology has been proposed as a viable remedy to tackle concerns surrounding data confidentiality due to its intrinsic attributes of immutability, verifiability, and decentralized structure. However, due to the limited capacity of the IoT, it might not be practical to run all the privacy preservation algorithms to ensure the privacy of the data on an IoT-supported blockchain. Thus, eliminating this complexity opens the door to attacks and compels us to

make security concessions. Furthermore, while the immutability of data is improved by the fact that all blocks in a blockchain contain identical information, this also raises privacy concerns for users.

*Transaction Efficiency:* There is a major issue with the transaction speed of a blockchain when integrating with IoT. It is universally acknowledged that IoT integration slows down systems. IoT systems can generate massive amounts of data promptly, while the pace of a blockchain may not be able to catch up. Popular blockchain systems, e.g., Bitcoin, can only process seven transactions per second. Ethereum, on the other hand, is faster, and can process around 30 transactions per second. Ethereum 2.0 has promised to increase the speed up to #100,000 transactions per second. On the other hand, several studies have addressed this issue, e.g., by helping to estimate the transaction throughput of the blockchain, increasing transaction throughput, etc. [139,140]. Nevertheless, the speed of transaction processing is still likely to be insufficient, especially in large-scale IoT applications.

*Law and policy:* Despite the blockchain method leading to numerous positive consequences for society, lawmakers and authorities have come under fire. Blockchain has run into a number of legal problems because of a lack of legal analysis [141]. A detailed understanding of blockchain processing will be essential to support the design and implementation of legal regulations governing blockchain activities.

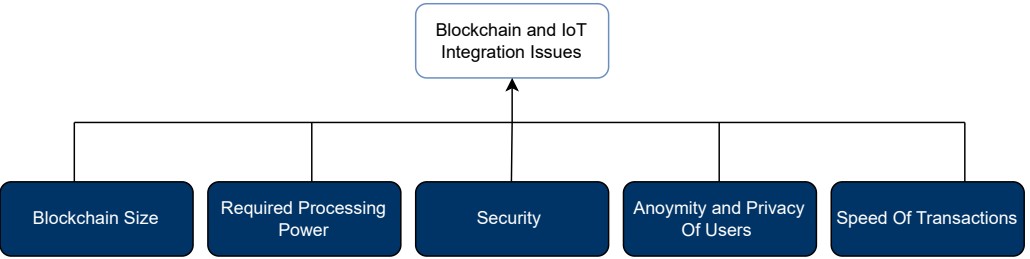

**Figure 8.** Issues associated with the integration of IoT and blockchain.

### 5.3. Blockchain and IoT Integration Strategies

Integrated blockchain–IoT architecture can be classified into three parts. The basic IoT model shown in Figure 9 is the simplest form of operation in which devices communicate with one another across a local area network. Since it does not directly utilize the complex computer methods of blockchain technology, this approach is typically the fastest to execute. Currently, blockchain's primary practical application is in the recording of digital exchanges and conversations. This architecture is best for uses in which a lesser level of security and faster communication are not crucial.

*IoT–Blockchain Model:* Blockchain is the sole means by which IoT gadgets can exchange data with one another. We can use the security properties of blockchain to protect IoT systems by implementing the architecture shown in Figure 9. Blockchain is used to capture and keep all of the IoT system's data and transactions, which can ensure the immutability of the data.

*Fog/Cloud Computing and Distributed Computing:* Many issues with the resource constraints of IoT devices have been alleviated thanks to advancements in cloud and fog computing [142], which form the basis of the IoT–blockchain model. Computing tasks such as hashing, encryption, decryption, etc., can be offloaded from IoT devices and onto fog and cloud infrastructure. Since blockchain necessitates a great deal of computing power, electricity, and related resources to function, the integration of fog/cloud computing with IoT technology becomes more relevant. IoT devices in the hybrid approach can either talk to blockchain nodes directly or to fog/cloud nodes indirectly. This allows us to take advantage of models combining the IoT with blockchain technology.

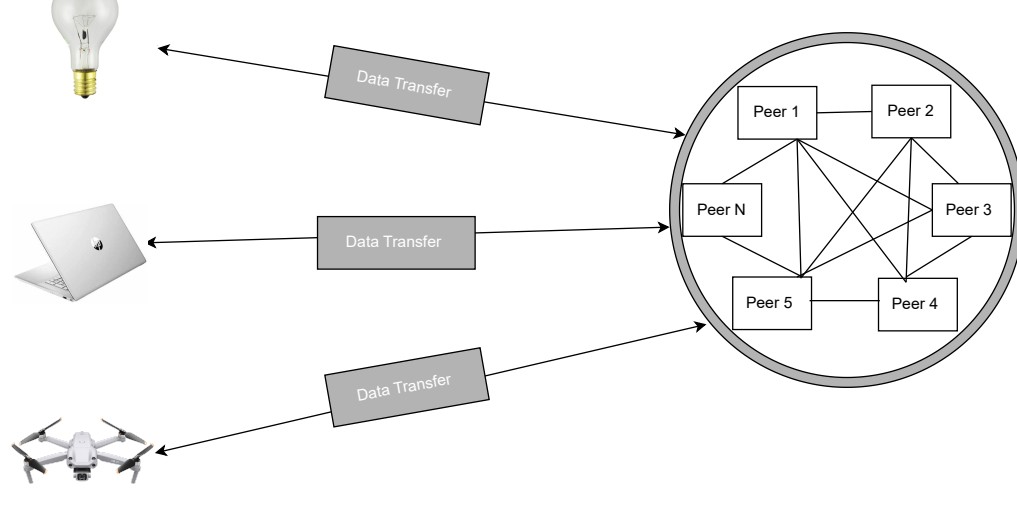

**IoT Devices**                                    **Blockchain Network**

**Figure 9.** Integration of IoT models.

## 6. Final Remarks

This study provides a thorough review of the advancements in security measures for the Internet of Things (IoT) by applying deep learning and blockchain techniques. Specifically, this study first elaborates on the fundamental characteristics of blockchain and deep learning and further provides an in-depth analysis of the advantages of their integration. This integration has been shown to improve data security and privacy in the current IoT while also enhancing the quality of service. However, we also realize that there are still many challenges associated with adopting blockchain and deep learning for secure IoT systems. Due to the limitations of the current IoT and defects of blockchain and deep learning, many solutions are still in their nascent stages, only representing proofs of concept. The survey results reported herein are expected to provide valuable insights for researchers to enhance their comprehension of the present research landscape and obstacles associated with the integration of blockchain and deep learning technologies in securing IoT systems.

**Author Contributions:** Conceptualization, A.F., Q.W., W.L. and W.Y.; methodology, A.F., Q.W., W.L. and W.Y.; writing—original draft preparation, A.F. and Q.W.; writing—-review and editing, Q.W., W.L. and W.Y.; problem space and formalization, A.F., Q.W., W.L. and W.Y.; supervision, Q.W., W.L. and W.Y.; project administration, Q.W., W.L. and W.Y. All authors have read and agreed to the published version of the manuscript.

**Funding:** This research received no external funding.

**Data Availability Statement:** Not applicable, the study does not report any data.

**Conflicts of Interest:** The authors declare no conflict of interest.

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
