# Peer review of "Survey of Distributed and Decentralized IoT Securities: Approaches Using Deep Learning and Blockchain Technology"

_futureinternet, doi:10.3390/fi15050178_

Round 1

Reviewer 1 Report

In this comprehensive survey, the authors explored the latest innovations in IoT security, including blockchain and deep learning. There are some errors interpreting the blockchain technologies and some typos, where I am not familiar with deep learning. The following points need to be clarified and corrected. More double checking on interpreting those technologies surveyed are expected to avoid endorsing those technologies. Many technologies are specific with some conditions. Please dont synthesize descriptions without mentioning the conditions.

1.      Figure 1, export à expert

2.      L 243, The blocks data essentially consists of a transaction log and a hash referencing the prior block. …” The consensus nonce is also essential. Or even more.

3.      L 252, with a specific number of leading zeros. …”, leading zero is just an example for description in Bitcoin white paper, it is actually comparing if the hash is less than target, otherwise the difficulty will be only changed by the power of 2.

4.      L 253, For block content to be immutable or final, Bitcoin is only probabilistic. After 100 blocks, it is only mature to be used in that the probability to be altered is very low, but still not final yet, never complete. However, public blockchain as this is trustful enough but slow and very energy consumptive. Even private blockchain is more efficient, the relied peers might change mind to become untrustful. Please do not trust private blockchain too much.

5.      Moreover, the content in public or private blockchains are mostly not encrypted and open to public or peers for transparency. Therefore, privacy is a difficult problem. Zero Knowledge or Homomorphic Encryption are very slow and not Turing complete to solve all privacy problems. Not permissioned blockchains can enhance privacy.

6.      L 266, The block header of Bitcoin does not have block size and number of transactions. They are in Block not block header. The current block hash is not encoded in the Merkel root field, it is the hash of block header.

7.      L 269, the trial counter is not nonce, note the extra nonce, and not checking the leading zero. The Merkel tree hashing is not employed frequently for every block.

8.      Target is not to specify the number of leading zero. Please find the exact meaning for Bitcoin. Especially, Ethereum is quite different.

9.      L 738, Secure Hash Algorithm (SHA), Advanced Encryption Standards (AES), RSA, etc. are based on the same set of common mathematical equations Are you sure the same set?

10.   What is HSM, AS,

11.   L 837, reason enough, reasons enough?

12.   L 880, Blockchain has solved the problem of data privacy on its network. Are you sure?

13.   L 901, Ethereum is much faster now.

14.   Without solving the fundamental finality problem in blockchain, many solutions are just proof of concept, let alone the performance and energy problems, especially the applications to deep learning. Privacy is another level of difficulty, especially for IoT. https://doi.org/10.3390/s23010015 addresses many of the problems above, but they are still a prototype under construction. Please dont mislead in your first half that many solutions are ready. In your final remarks, I still dont see that the a number of issues and necessary restrictions are significant to the realization of blockchain plus deep learning.

Reviewer 2 Report

Review comments regarding assigned manuscript entitled “Survey on Distributed and Decentralized IoT Securities: Approaches Using Deep Learning and Blockchain Technology” given below:

1.        Some recent literatures can be added to improve the theoretical depth of this paper:

1.        Hang, L., Ullah, I., Yang, J., & Chen, C. (2022). An improved Kalman filter using ANN-based learning module to predict transaction throughput of blockchain network in clinical trials. Peer-to-Peer Networking and Applications, 1-18.

2.        The authors should improve the quality of figures throughout the paper.

3.        The English editing should be improved as there are some typos and grammar mistakes.

Round 2

Reviewer 1 Report

There are still wrong concept and description of blockchain. Please correct again.

1. In reply of question 4, “The information contained in a block is normally considered to be immutable once consensusbased validation and verification of transactions have been completed..."

   Since the data is hash-chained, it is immutable or very difficult to change anyway. Every node might decide to accept different blocks so that there might have branches of blockchain and reaching consensus is needed. Reaching consensus is not a process to discuss with other nodes. It is naturally done individually and independently because the longest chain policy. The problem is when to be “completed”. The immutable data might be replaced by those from new longest chain. If the chain is never completed, the data seen by some nodes will be replaced someday. In Bitcoin, it is not completed or final in a definite time. It is just suggested 6 confirmations or blocks is OK because the probability to be replaced is low enough but 100 blocks to be mature to spend the coins for save. This is because it can randomly trust a node to provide trustless trust in public blockchain as Bitcoin, provided that time will find the longest and trustful chain eventually if the malicious nodes are not too many. But it is slow… On the other hand, people propose private blockchain. The permissioned nodes are trustful and they communicated to reach consensus. It is fast but not really trustful because we need to count on that those trustful nodes do not change mind. “However, despite the public or private blockchain, the peer users in the system are the core components to ensure data security. The large group of collusive malicious peer users is always able to lead the system to be untrustful.is not clear. Actually, most current solutions using private blockchain is not trustful not only because trustful nodes might become malicious, the unintentional wrong immutable data might be worse. Please do not mislead that current private blockchains are trustful.

2. From reply of question 3, 6, 7, please read again the format of block header and how mining is conducted.

The proof of work is just a piece of information to proof some enough work remotely and it can be checked easily locally. In Bitcoin it is to check that the hash of block header is less than target. Not a puzzle, no number of leading zeros. The hash is the block ID or index. The target is a function of nBit. nBit can be seen as a compact float point representation of target. Target is reverse of difficulty. nBit and block version are fields of block header. Merkel root is a hash of all transactions including the basecoin transaction which contains a piece of information called extra nonce. Mining is to find nonce and the extra nonce satisfying the inequality. The frequency of mining a block is never changed and never defined. The difficulty is adjusted every 2016 blocks so that the average of mining a block in next 2016 blocks is close to 10 minutes as possible.

3. Ethereum is claimed to reach 1000-2000 TPS.

Round 3

Reviewer 1 Report

Please know how the blockchain works by its source code, not only from web or papers. The following needs to be checked and corrected. I have mentioned in previous reviews but not corrected.

1. The is no such thing as comparing the leading zeros of hashes in the source code for Bitcoin mining. It is not a complicated math problem or puzzle. It is just a simple inequality test of hash(block header) < target. The leading zero is just a just-like description of the idea from white paper of Bitcoin, but not true exactly and spread for years. We need to compare the 0's and 1's after the first 1 (or not zero) to test the inequality, not just count the leading zeros. Otherwise, the difficulty change would be a difference of to the power of 2.

2. The block ID is the hash of the block header, not encoded in merkel root. The merkel root is the root of hash tree of all transactions at leaves, representing the abstract of all transactions.
